engineering geology/energy

dilatancy behaviour, permeability evolution, crack density, unloading stress path

**Author for correspondence:**
Chao Liu
e-mail: liuc_rm@cumt.edu.cn

# Dilatancy behaviour and permeability evolution of sandstone subjected to initial confining pressures and unloading rates

Honggang Zhao[1,2], Chao Liu[1,2,3,4] and Gun Huang[1,2]

[1]State Key Laboratory of Coal Mine Disaster Dynamics and Control, and [2]School of Resources and Safety Engineering, Chongqing University, Chongqing 400030, People's Republic of China
[3]State Key Laboratory for Geomechanics and Deep Underground Engineering, and [4]School of Mechanics and Civil Engineering, China University of Mining and Technology, Xuzhou 221116, People's Republic of China

 HZ, 0000-0001-5086-1325

Mechanical response, deformation behaviour and permeability evolution of surrounding rock under unloading conditions are of significant importance in rock engineering activities. In this research, triaxial experiments of sandstone subjected to different initial confining pressures and unloading rates under fixed axial stress were conducted. The results showed that sandstones experienced shear dilatancy before failure. However, the dilatancy factor did not decrease with increasing confining pressure, i.e. the dilatancy behaviour was not suppressed, which contradicted the phenomenon under increasing axial stress. The crack density also increased with increasing initial confining pressure. Furthermore, the normalized permeability was positively correlated with unloading rates. The sandstone permeability was closely related to the shear dilatancy behaviour. In the accelerated dilatancy stage, the relationship between normalized permeability and volumetric strain was linear at low unloading rates and nonlinear at high unloading rates. The linear/nonlinear relationship between them can directly affect the temporality of respective mutation, so as to guide the prevention of geological disasters at different excavations rates.

## 1. Introduction

Unloading behaviour of rock mass is inevitable in rock engineering activities, such as tunnel/underground roadway/

underground chamber excavation and failure of supporting structure. On one hand, unloading of rock mass may cause major engineering disasters, such as rock burst [1,2]; on the other hand, through protective layer mining technology, gas outburst in protected coal seam may be eliminated by *in situ* stress release and transfer [3].

As the confining pressure on the working face and its adjacent area are small, it will inevitably cause the volume expansion of the surrounding rock, and a large rock displacement will be generated along the direction of the free surface. Li *et al*. [1] considered that the propagation of a large number of fractures and accelerated deformation in surrounding rocks were the precursory characteristics of rock burst. Underground supporting was to actively control the overexpansion deformation of surrounding rock caused by excavation with an appropriately high supporting force. The dilatancy was an important mechanical property of rock, and the failure of rock mass was closely related to its dilatancy mechanism [4].

Dilatancy is the nonlinear increase of volume caused by internal crack compaction, initiation, propagating and shear slip inside rock mass. The dilatancy behaviour appears from the beginning of the yield stage to the strain softening stage. It is mainly dependent on the plastic strain and confining pressure [5,6], which is consistent with the mechanical response of the surrounding rock near the excavation boundary of the underground engineering [7]. Mahmutoglu *et al*. investigated the correlation between the post-peak strength and the non-elastic volumetric strain of fractured marble under low confining pressure. It was noted that the dilatancy curve of post-peak can be replaced by a straight line, and the slope of the curve decreased with the increase of confining pressure [8]. Jeng *et al*. concluded that the sandstone exhibited plastic strain before it reached the yield failure and induced obvious dilatancy deformation. The results showed that the shear dilatancy caused a great displacement of the soft rock [9]. Similarly, plastic behaviour of rocks was not uncommon especially for shales and coal-rich layers but less so for limestone and sandstone [10,11]. Roche & van der Baan concluded that the anisotropic Young's moduli tended to reverse the effect of plasticity strain, thus decreasing the likelihood of failure in the shales and coals [12]. These studies show that sandstone has obvious dilatancy behaviour. Shao *et al*. [13] proposed that the damage was caused by volume expansion induced by the development of microcracks. Salari *et al*. considered that the elastic damage was caused by the expansion of the volume. The energy release rate was characterized by elastic volumetric strain and plastic volumetric strain, which could represent the damage variable and strength softening model of rock [14]. In fact, the dilatancy angle of rock mass was related to confining pressure and plastic shear strain, and is not fixed or linearly changed [6].

In underground rock engineering, extremely complex geological conditions may be encountered. Gas outburst accident occurred in Dafang Tunnel in Bijie City, Northwest Guizhou Province, on 2 May 2017, because of the existence of karst, fault-fractured zone and mine waste in the tunnel. The flow field cannot be ignored, and the dilatancy behaviour has an important influence on the permeability evolution of rock. Yin *et al*. [3,15] considered that the change of permeability was closely related to the change of volumetric strain under unloading confining pressure, leading to a rapid increase stage of permeability. Lee *et al*. performed the shear seepage test of granite. It was found that shear effect had an enormous influence on the fracture permeability, and a smaller shear displacement will lead to a significant increase in fracture permeability [16].

Furthermore, the loading and unloading rates are also closely related to the deformation behaviour, strength and permeability evolution of rock mass. Zhao *et al*. considered that the violence of rock mass in the process of failure and the release of acoustic emission energy were related to the unloading rate. With the decrease of unloading rate, the failure mode changed from strainburst to spalling [2]. Sangha and Dhir considered that shear sliding was the main failure mode at a high loading rate. With the further decrease of loading rate, the failure modes were found to be more inconsistent and the strength did not change significantly [17]. Huang & Li, however, concluded that with the increase of the unloading rate, the failure mode of rock gradually changed from shear to tensile [18]. Generally, a rapid excavation speed will lead to a high loading–unloading rate of rock mass near the excavation boundary [2], and the mining methods with very large mining intensity, such as fully mechanized top-coal caving, have the characteristics of fast advanced speed. A high mining rate might make the stress redistribution uniformity inferior, resulting in an unstable energy release, thereby causing geological disasters [19]

Previous studies have rarely linked rock dilatancy behaviour and permeability mutation characteristics under unloading stress paths. Therefore, the triaxial mechanics and seepage experiments of sandstone with different combinations of initial confining pressure and unloading

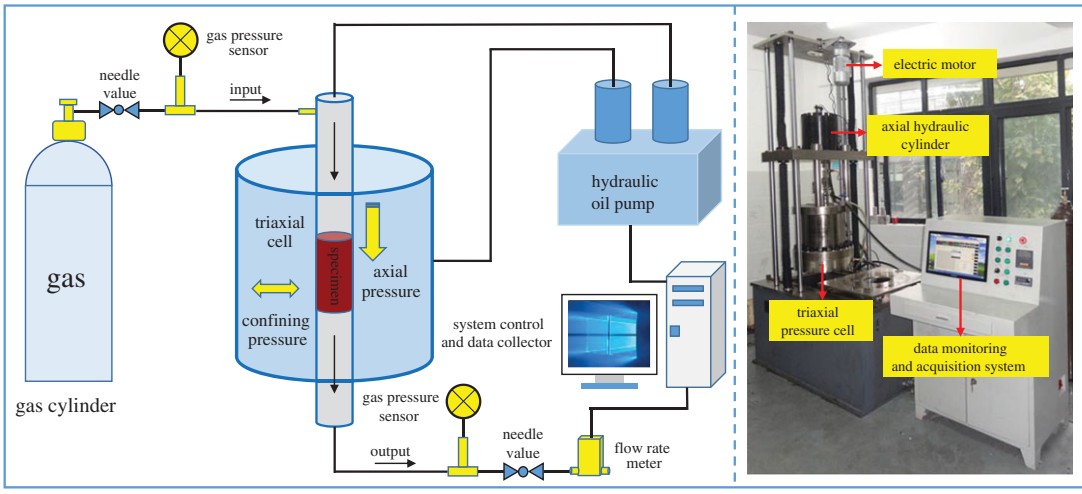

**Figure 1.** Schematic and physical diagram of the apparatus.

confining pressure rate (UCT) under fixed axial stress are performed in order to investigate the dilatancy behaviour, permeability evolution and their relationship.

# 2. Experimental apparatus and scheme

## 2.1. Experimental apparatus

The experiment was performed with the servo-controlled seepage apparatus for thermal–hydrological–mechanical (THM) coupling of coals and rocks [20], as shown in figure 1. The apparatus is composed of a main engine, a hydraulic power source, a constant temperature oil bath, and a measurement and control system. Specifically, the main engine is composed of an axial stress loading cylinder, a triaxial pressure cell and a lifting mechanism. The measurement and control system are composed of an axial loading system, a confining pressure loading system, a temperature control system, a pneumatic control system, an electro-hydraulic proportional control system, a data measurement and acquisition system and an auxiliary system. The apparatus can be used to perform coal and rock experiments under various stress paths and engineering conditions, such as loading and unloading experiments of coals and rocks under the coupling action of stress field, flow field and temperature field, and a hydraulic fracturing experiment.

## 2.2. Experimental specimens

Experimental specimens were obtained from an outcrop of feldspar sandstone in Chayuan New Area, Chongqing. The sandstone is cut, coarsely ground and polished, and processed into experimental specimens with a diameter of 50 mm and a height of 100 mm (figure 2). The relative non-parallelism error between the faces of opposite ends did not exceed 0.02 mm. The uniaxial compressive strength (UCS) of sandstone is 46.90 MPa. The porosity of sandstone measured by the mercury intrusion method is 4.15%, and the micropore is well developed.

## 2.3. Experimental scheme

In the paper, the effects of UCTs on the deformation behaviour and permeability evolution of sandstone under fixed axial stress were investigated. We performed a conventional triaxial compression (CTC) test to determine the axial stress level under the UCT. The details are as follows: *Step* 1—Axial stress and confining pressure are applied to a hydrostatic pressure state of 10, 15, 20 and 25 MPa, at an identical rate of 0.05 MPa s$^{-1}$. *Step* 2—The gas inlet valve is opened, and a gas pressure of 3 MPa is applied to sandstone specimens under different confining pressures. *Step* 3—When the flow rate is stable, the axial stress is applied at the rate of 0.01 mm min$^{-1}$ until the specimen fails.

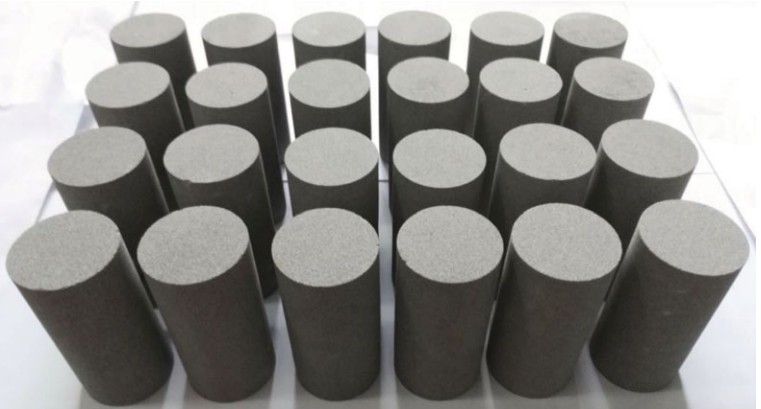

**Figure 2.** Sandstone specimens.

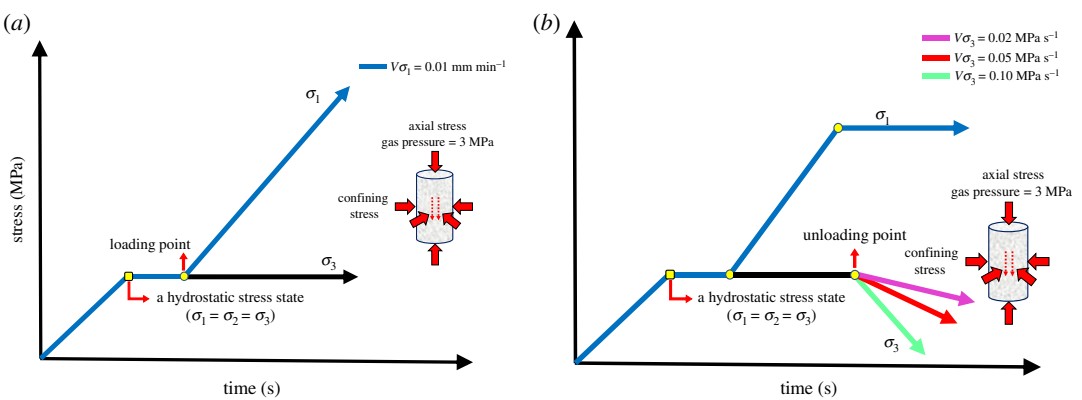

**Figure 3.** Schematic diagram of the experimental scheme: (*a*) conventional triaxial compression test and (*b*) unloading confining pressure test.

Subsequently, we conducted the UCT experiments. The details are as follows: *Step* 1—Axial stress and confining pressure are applied to a hydrostatic pressure state of 15, 20 and 30 MPa, at the identical rate of 0.05 MPa s$^{-1}$. *Step* 2—The gas inlet valve is opened, and a gas pressure of 3 MPa is applied to the sandstone specimens under different confining pressures. *Step* 3—When the flow rate is stable, the axial stress is applied up to approximately 70% of peak stress measured by the CTC experiment at a rate of 0.05 MPa s$^{-1}$. *Step* 4—The axial stress is kept constant, and the unloading confining pressure experiment is performed at the rate of 0.02, 0.05 and 0.1 MPa s$^{-1}$. *Step* 5—Once the specimen fails, the force control is switched to the displacement control immediately, and the loading is continued at the rate of 0.1 mm min$^{-1}$ until the residual strength is stable.

The specific stress paths are shown in figure 3, and the conventional triaxial strengths of sandstone specimens under different confining pressures with 3 MPa gas pressure are shown in figure 4.

# 3. Stress–strain relationship

In the paper, the third step of the UCT experiment is taken as the starting point of the study, and the deformation behaviour and permeability evolution of sandstone are analysed. The deviator stress–strain curves of sandstone are shown in figure 5. The deformation trend was consistent under different experimental conditions. The axial stress remained unchanged and the confining pressure is continuously reduced, so that the deviator stress ($\sigma_1 - \sigma_3$) was continuously increased. Therefore, the stress path can also be regarded as a deviator stress loading path. The volume expansion rate of the sandstone samples was accelerated by the rapid increase of hoop strain compared with the axial strain. The strength and volumetric strain of sandstone increased with the increase of confining pressure. Table 1 lists the deviator stress and strain at the time of failure.

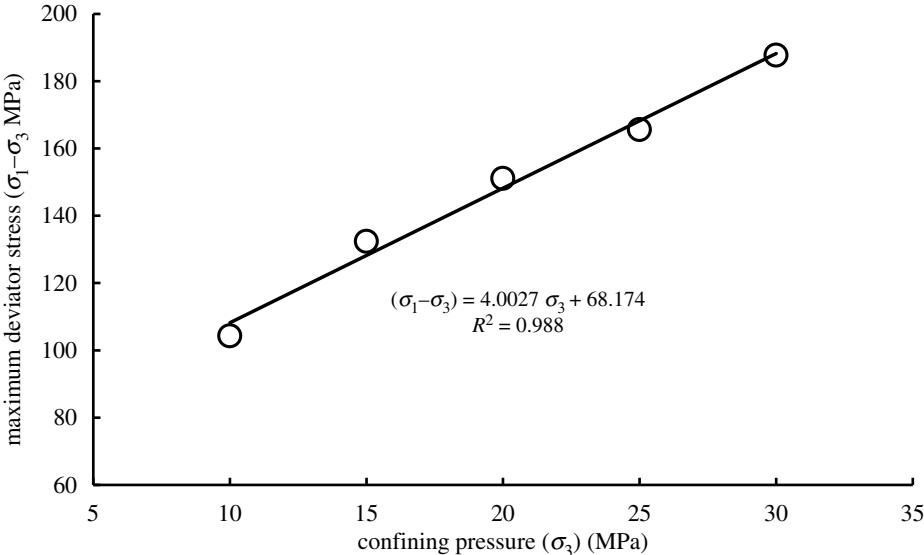

**Figure 4.** Relationship between maximum deviator stress and confining pressure under the conventional triaxial compression test.

## 3.1. Dilatancy behaviour

The sandstone formed an inclined failure plane at an angle to the axial stress, and some powder was found on the failure plane. To investigate whether these powders were caused by shear or tensile, we performed a Brazilian split test and a direct shear test on the sandstone to determine the characteristics of the failure plane in two extreme cases. The dispersion of sandstone particles (including debris) is shown in figure 6. We found that the sandstone was partially dispersed under shear effect, and more particles (including debris) fell on the table. It was a typical surface spalling or grain detachment, which is consistent with previous studies [21,22]. The main feature of sandstone failure due to tensile stress was the occurrence of intergranular cracks, with only a small number of scattered particles. This was also in line with some previous research results [21–23]. In addition, the failure contour caused by the shear effect was more curved than that caused by the tensile effect (figure 6). Shimizu et al. [24] considered that the shear crack is stepped. In view of the above analysis, we concluded that the failure of sandstone was mainly caused by the shear effect in the experiment.

Furthermore, lower confining pressure results in distinct brittle drop of sandstone, and no brittle-ductile transitional behaviour was observed as shown in figure 5. The phenomenon means that the shear dilatancy (or climbing effect [25]) and shear localization of the sandstone are not suppressed, in which the deformation behaviour tends to be non-uniformly distributed [26]. These microcracks distributed within the damage zone of the shear fracture may not be observed with the naked eyes due to the fact that the length of microcracks is generally of a length comparable to the sandstone grain size [21].

The internal friction angle affects the shear dilatancy. Figure 7 shows the change of the internal friction angle of sandstone in the case of $v = 0.02\,\mathrm{MPa\,s^{-1}}$. It can be seen that the internal friction angles are substantially identical, indicating that it has the same effect on the shear dilatancy, and the dilatancy behaviour of sandstone is significantly affected by the applied stress. The shear effect increases with increasing deviator stress, which promotes the sliding of closed cracks, and the opening of microcracks at the asperities. To summarize, the high deviator stress causes a large volumetric strain increment (table 1), i.e. the sandstone has experienced significant shear dilatancy.

In shear motion, the strain was large enough to lead the shear band to dilate [27]. To describe the shear dilatancy of rock caused by the growth of microcracks or the staggered uplift of fracture surfaces (including grains) due to deviator stress, the dilatancy factor (DF) can be used to characterize the shear localization behaviour [26,28,29]

$$\mathrm{DF} = -\sqrt{3}\,\frac{\Delta\varepsilon_{V,P}/\Delta\varepsilon_{a,P}}{3 - \Delta\varepsilon_{V,P}/\Delta\varepsilon_{a,P}}, \tag{3.1}$$

$$\Delta\varepsilon_a = \Delta\varepsilon_{a,e} + \Delta\varepsilon_{a,P}, \tag{3.2}$$

$$\Delta\varepsilon_r = \Delta\varepsilon_{a,r} + \Delta\varepsilon_{a,r} \tag{3.3}$$

and

$$\Delta\varepsilon_V = \Delta\varepsilon_{V,e} + \Delta\varepsilon_{V,P}, \tag{3.4}$$

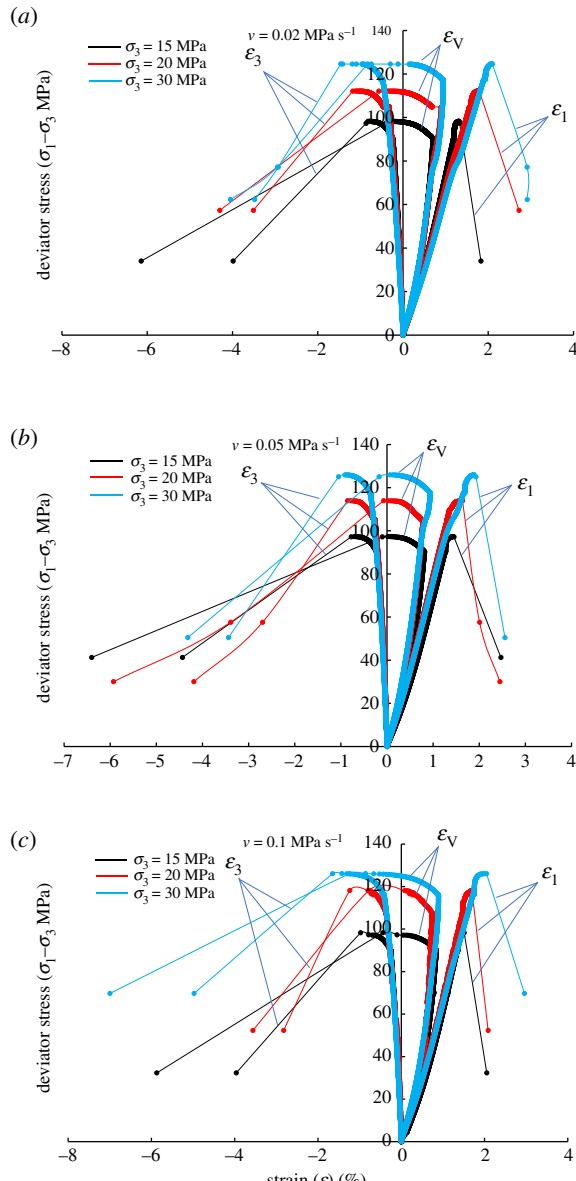

**Figure 5.** Stress–strain relationship of sandstone under different unloading rates. (*a*) $v = 0.02$ MPa s$^{-1}$, (*b*) $v = 0.05$ MPa s$^{-1}$ and (*c*) $v = 0.1$ MPa s$^{-1}$.

**Table 1.** Deviator stress and strain of the sandstone specimens at the peak stress.

| $v$ (MPa s$^{-1}$) | $\sigma_3$ (MPa) | $(\sigma_1-\sigma_3)_{peak}$ (MPa) | $\varepsilon_{vpeak}$ (%) | $\varepsilon_{1peak}$ (%) | $\varepsilon_{3peak}$ (%) | $\varepsilon_{vmax}$ (%) | DF | $\gamma$ | $\gamma'$ |
|---|---|---|---|---|---|---|---|---|---|
| 0.02 | 15 | 97.31 | −0.403 | 1.33 | −0.867 | 0.709 | 1.056 | 2.19 | 0.657 |
| | 20 | 112.10 | −0.569 | 1.792 | −1.18 | 0.905 | 1.019 | 3.12 | 0.757 |
| | 30 | 124.6 | −0.829 | 2.095 | −1.462 | 0.943 | 1.29 | 3.53 | 0.793 |
| 0.05 | 15 | 97.18 | 0.0131 | 1.431 | −0.709 | 0.815 | 0.82 | 2.61 | 0.706 |
| | 20 | 113.93 | −0.0733 | 1.636 | −0.855 | 0.782 | 0.595 | 2.64 | 0.71 |
| | 30 | 125.03 | −0.172 | 1.928 | −1.05 | 0.938 | 0.834 | 3.04 | 0.75 |
| 0.1 | 15 | 98.25 | −0.441 | 1.511 | −0.976 | 0.839 | 1.208 | 2.94 | 0.74 |
| | 20 | 118.17 | −0.778 | 1.696 | −1.237 | 0.751 | 1.107 | 2.66 | 0.712 |
| | 30 | 126.05 | −1.251 | 2.047 | −1.649 | 0.896 | 1.121 | 3.33 | 0.776 |

(a)

(b)

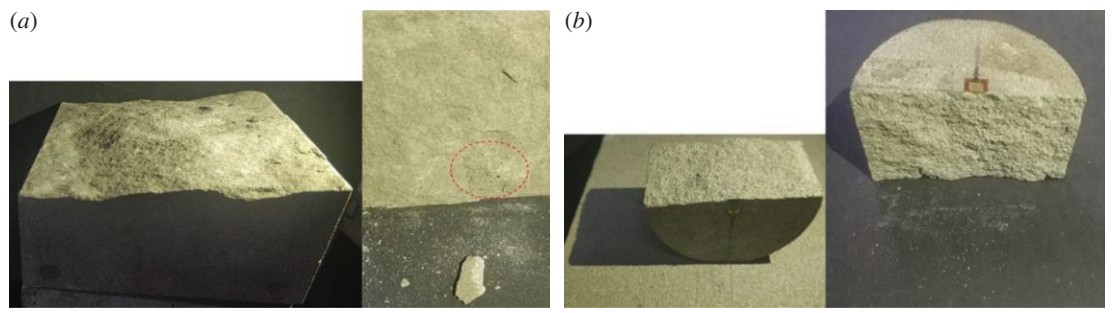

**Figure 6.** Failure morphology of sandstone. (a) Failure morphology of sandstone under the direct shear test. (b) Failure morphology of sandstone under the Brazilian split test.

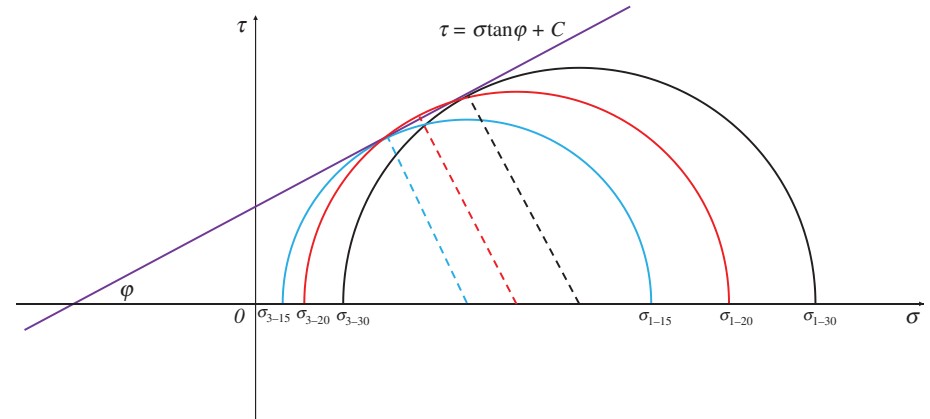

**Figure 7.** Internal friction angle of sandstone under three initial confining pressure conditions in the case of $v = 0.02$ MPa s$^{-1}$. $\sigma_{1-30}$, $\sigma_{1-20}$ and $\sigma_{1-15}$ are the peak stresses of sandstone under the initial confining pressures of 30, 20 and 15 MPa, respectively. $\sigma_{3-30}$, $\sigma_{3-20}$ and $\sigma_{3-15}$ are the confining pressures corresponding to the sandstone in $\sigma_{1-30}$, $\sigma_{1-20}$ and $\sigma_{1-15}$, respectively.

where DF is the ratio of plastic strain increment. $\Delta\varepsilon_{a,e}$ and $\Delta\varepsilon_{a,p}$ are elastic and plastic axial strain increments, respectively. $\Delta\varepsilon_{V,e}$ and $\Delta\varepsilon_{V,p}$ are elastic and plastic volumetric strain increments, respectively.

The calculated results are shown in table 1. What interests us is that DF has two distinct characteristics. One is that DF is larger than that in the related literature [26,29], indicating that $\Delta\varepsilon_{V,p}/\Delta\varepsilon_{a,p}$ is larger, i.e. $\Delta\varepsilon_{V,p}$ is larger than $\Delta\varepsilon_{a,p}$. This implies that sandstone dilatancy is very significant in the experiment. The other is that $DF$ does not decrease with the increasing confining pressure, and the decrease of DF means that the shear dilatancy is suppressed. The authors believe that the above two characteristics are related to the mechanical properties and stress state of sandstone. The strength of the sandstone used in our research is greater than that in the related literature [26,29], and the experimental conditions do not meet the stress regime of the brittle–ductile transition of rock, and there is no cataclastic flow [29,30].

In addition, we studied the dilatancy behaviour of sandstone under the stress path of unloading confining pressure rather than the loading axial stress [26,29]. There are significant differences in the deformation behaviour of sandstone caused by the stress path between the two. For example, the failure of the surrounding rock of the tunnel induced by shear-dilatancy comes from the unloading dilatancy, and the dilatancy failure in the triaxial compression test comes from loading failure. The inducement of dilatancy behaviour is different between the two. In a brittle state, high confining pressure is conducive to fracture compaction, which means natural or pre-existing closed cracks increase. Closed cracks are the necessary condition for shear sliding [31], and coupled with high deviator stress, resulting in that the shear dilatancy of sandstone is significant under $\sigma_3 = 30$ MPa. It is worth noting that as the initial confining pressure increases, DF does not increase regularly, which may be related to the discreteness of sandstone, but it is not the concern of this paper. We focus on whether the dilatancy behaviour is significantly suppressed by the increase of initial confining pressure during unloading process.

Furthermore, we investigated the evolution of the crack density parameter of sandstone. The sliding crack model can better characterize the inelastic mechanical behaviour of rock using effective Young's modulus [31]. When the normalized stress $\hat{\sigma}(= \sigma/\sigma c) < 1$, the open cracks inside sandstone are always

in the open state. When $\hat{\sigma} > 1$ and the cracks are at a certain angle to the axial stress (or confining pressure), the cracks close and slip as the applied stress increases. The opening, closing and sliding of cracks affect the compliance of sandstone. However, only the closed cracks can be stimulated to form a shear slip. Therefore, we only consider the situation in which $\hat{\sigma} > 1$. In the uniaxial compression experiment, the increase of the axial stress is equivalent to the increase of the deviator stress (in the special case of $\sigma_3 = 0$). The deviator stress also increases under the condition of unloading confining pressure. In addition, rocks exhibit different mechanical responses under loading and unloading stress paths; however, from the viewpoint of Mohr stress circle, only the stress point moves to the left or right. Therefore, it is reasonable to use the sliding crack model. The key equations are as follows:

$$(\sigma_1 - \sigma_3) = E_{\text{eff}} \varepsilon_a, \tag{3.5}$$

$$\hat{E} = \frac{E_{\text{eff}}}{E}, \tag{3.6}$$

$$\hat{\sigma} = \frac{(\sigma_1 - \sigma_3)}{\sigma_c} = \frac{2(\sigma_1 - \sigma_3)}{E\alpha}, \tag{3.7}$$

$$\frac{1}{\hat{E}} = 1 + c_{\text{open}} + c_{\text{sliding}}, \tag{3.8}$$

$$c_{\text{open}} = 2\gamma \left( \arcsin \sqrt{\frac{1}{\hat{\sigma}}} - \frac{\sqrt{\hat{\sigma} - 1}}{\hat{\sigma}} \right), \tag{3.9}$$

$$c_{\text{sliding}} = \gamma \left[ \frac{1}{2} \left( \beta - \frac{\sin 4\beta}{4} \right) - \mu \sin^4 \beta - \frac{\mu}{\hat{\sigma}} \cos 2\beta \right]_{\beta_c}^{\beta_s}, \tag{3.10}$$

$$\beta_c(\hat{\sigma}) = \arcsin \sqrt{\frac{1}{\hat{\sigma}}} \tag{3.11}$$

and

$$\beta_s(\hat{\sigma}) = \frac{1}{2} \left[ \arccos \frac{\mu \left( 1 - \frac{2}{\hat{\sigma}} \right)}{\sqrt{1 + \mu^2}} + \arctan \frac{1}{\mu} \right], \tag{3.12}$$

where $\sigma$ is applied stress in MPa. $\sigma_c$ is normal stress required for crack closure and contact in MPa. $(\sigma_1 - \sigma_3)$ is deviator stress in MPa. $\varepsilon_a$ is axial strain. $\hat{E}$ is normalized Young's modulus in MPa. $E$ is Young's modulus of sandstone under triaxial stress regime in MPa. $E_{\text{eff}}$ is effective Young's modulus in MPa. $\alpha$ is an initial aspect ratio of the crack. In the paper, it was considered that $\alpha$ was the ratio of the minor axis to the major axis of an elliptical crack in sandstone. Similarly, Zhang & Bentley [32] rationally believed that the cracks in sandstone were also elliptical. $c_{\text{open}}$ is the compliance contributed by open cracks to sandstone. $c_{\text{sliding}}$ is the compliance contributed by sliding cracks to sandstone. $\gamma$ is crack density parameter. $\beta$ is the angle between the major axis of the crack and the axial stress, and is less than 90°. $\beta_s$ is the critical angle of crack sliding. $\beta_c$ is the critical angle of crack closure. $\mu$ is the friction coefficient of the relative surface of the crack.

Assume $\mu = 0.25$, $E = 15$ GPa and $\alpha = 0.001$ to calculate the crack density parameter. The deviator stress–axial strain fitting represented by figure 8b is performed in the unloading confining pressure process. As shown in figure 9, the model matched well with the experimental data, especially the inelastic mechanical behaviour of the accelerated dilatancy stage. The crack density parameter increased with the increase of initial confining pressure or peak deviator stress (table 1), which was consistent with the change of volumetric strain. The large crack density parameter was caused by the most serious defect of the sliding crack model. That was, the model assumed that there was no interaction between cracks inside sandstone. Cracks propagated and slipped independently. The assumption was unrealistic, such as the widespread stress shadow in hydraulic fracturing [33,34]. Considering the interaction between cracks, Zimmerman [35] used differential effective medium theory to obtain a more realistic crack density parameter ($\gamma'$)

$$\gamma' = \frac{1}{\pi} \ln (1 + \pi\gamma). \tag{3.13}$$

The evolution of the crack density parameter under applied stress was discussed. However, the exploration of its true value was beyond the scope of our study.

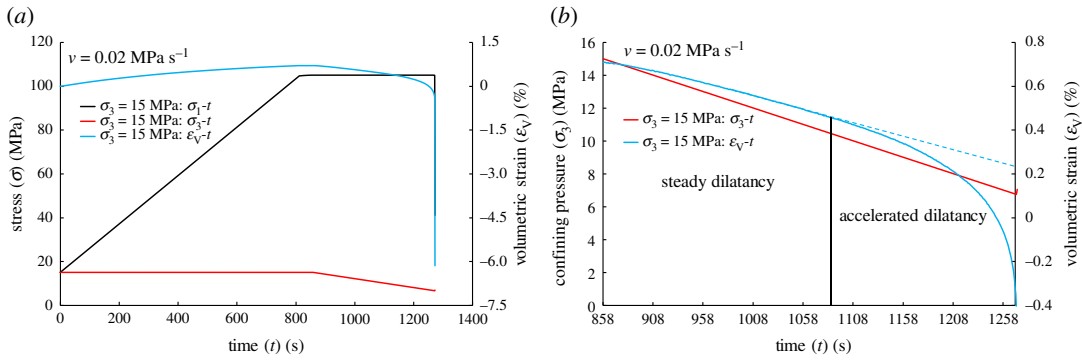

**Figure 8.** (a) Axial stress ($\sigma_1$), confining pressure ($\sigma_3$) and volumetric strain ($\varepsilon_V$) versus time ($t$) in the case of $v = 0.02$ MPa s$^{-1}$. (b) Relationship between confining pressure ($\sigma_3$), volumetric strain ($\varepsilon_V$) and time ($t$) during the unloading confining pressure stage in the case of $v = 0.02$ MPa s$^{-1}$. According to the evolution of volumetric strain, it can be divided into the steady dilatancy stage and the accelerated dilatancy stage.

## 3.2. Axial stress plateau

Table 2 shows the duration of stress plateau under different initial confining pressures and unloading rates. The failure of sandstone occurred only after the axial stress reaches the target value for a period of time, and then failure was instantaneous (figure 8a). The failure process of sandstone can be regarded as two stages: the stress plateau stage and the failure instability stage. From the stress plateau stage, the greater unloading rate resulted in the more vulnerable sandstone, which may be disadvantageous in engineering [18]. Similarly, it can be found from the deformation behaviour of sandstone that the volumetric strain changes steadily in the early stage of stress plateau (figure 8b). Although new microcracks have begun to be produced inside the sandstone at dilatancy stage, the dilatancy behaviour of sandstone includes elastic volume recovery, and the sandstone still exhibits considerable bearing capacity. Therefore, the strain develops steadily at the initial stage of unloading.

In the later stage of stress plateau, the strain increases sharply with continuous unloading, and the behaviour of accelerated dilatancy occurs. The uncontrolled development of fractures is essentially the reason that the axial stress is getting closer to the ultimate bearing capacity of sandstone. In addition, at the later stage of stress plateau, higher rates resulted in the faster strain change.

There are increasing transgranular cracks in the shear damage zone, and the orientation along the axial stress direction is more preferential as unloading rate increases [21], especially due to the low porosity of the sandstone used in the experiment, which is easier to capture [36,37]. The shear cracks tend to propagate and develop after the tensile crack is formed [36–40]. With the continuous unloading of confining pressure, higher deviator stress inhibits the initiation and development of tensile cracks [40], which is more conducive to slipping of the previously formed shear cracks. The unloading rate may have a threshold associated with the macroscopic failure mode to distinguish between shear-dominated failure and tensile-dominated failure. Huang and Li performed triaxial compression experiments on marble at different unloading rates and concluded that the failure mode gradually changed from shear failure to tensile failure with the increase of unloading rates [18]. In our experiment, the unloading rate is relatively small, and the shear failure is dominant, which may be related to the rock properties and loading conditions [21,38]. Shear failure means more microseisms [41]. When high excavation speed cannot be avoided, it is necessary to monitor the location of the natural flaws in front of the working surface or the failure planes induced by stress perturbation to prevent the release of a large amount of shear strain energy due to faults or defects slippage [42].

## 3.3. Prediction of failure time

Similarly, the unloading stress path of sandstone is similar to the pressurization or decompression behaviour of volcanic activity [43]. The failure and instability of sandstone are also the result of cumulative damage. We can use the concept of time-to-failure proposed by Voight to predict sandstone failure [44]

$$\frac{\mathrm{d}^2\Omega}{\mathrm{d}t^2} = A\left(\frac{\mathrm{d}\Omega}{\mathrm{d}t}\right)^\chi, \tag{3.14}$$

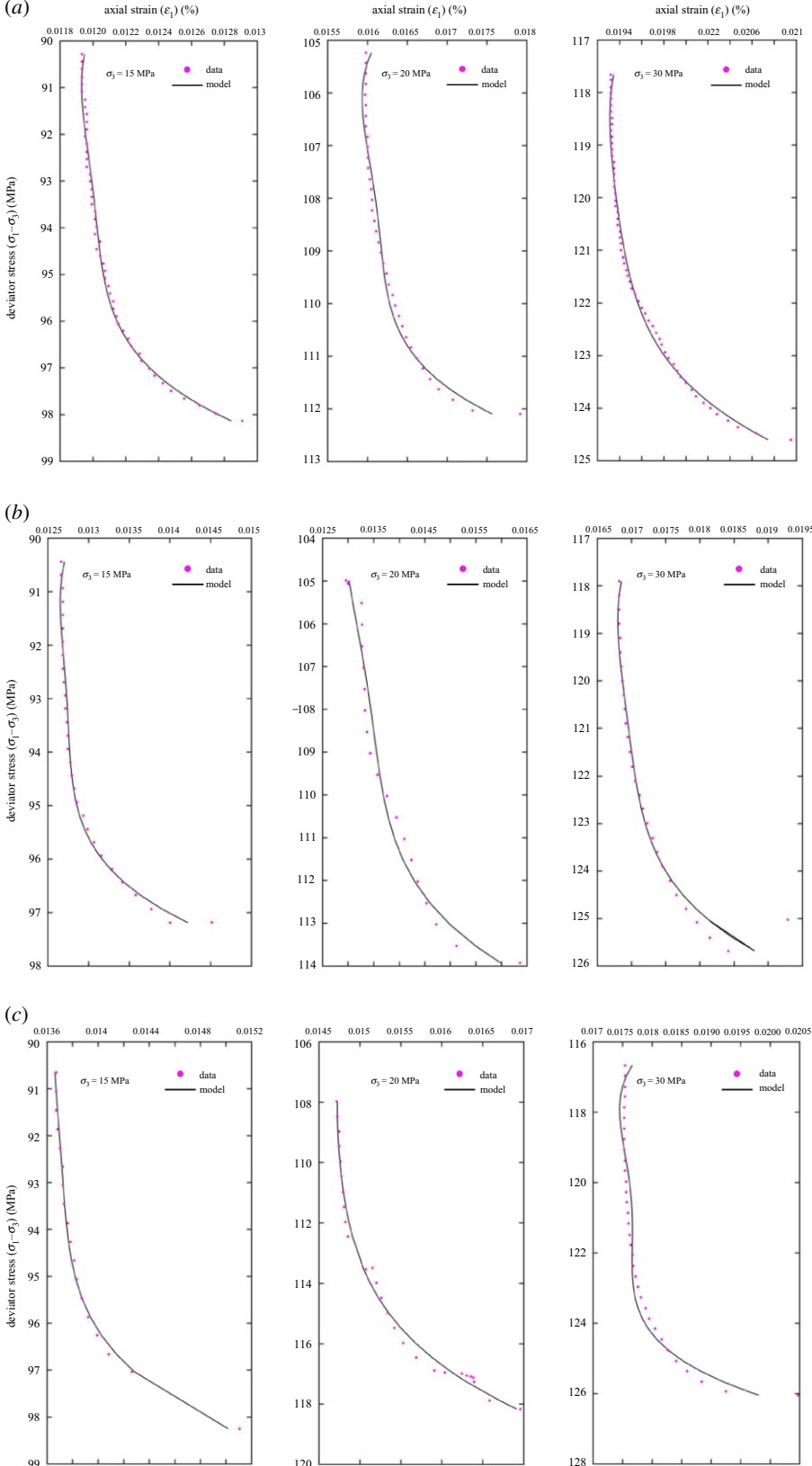

**Figure 9.** The sliding crack model was used to fit the deviator stress $(\sigma_1-\sigma_3)$ and axial strain $(\varepsilon_1)$ in the unloading confining pressure stage and to compare with the experimental results under different confining pressures and unloading rates. (*a*) $v = 0.02$ MPa s$^{-1}$, (*b*) $v = 0.05$ MPa s$^{-1}$ and (*c*) $v = 0.1$ MPa s$^{-1}$.

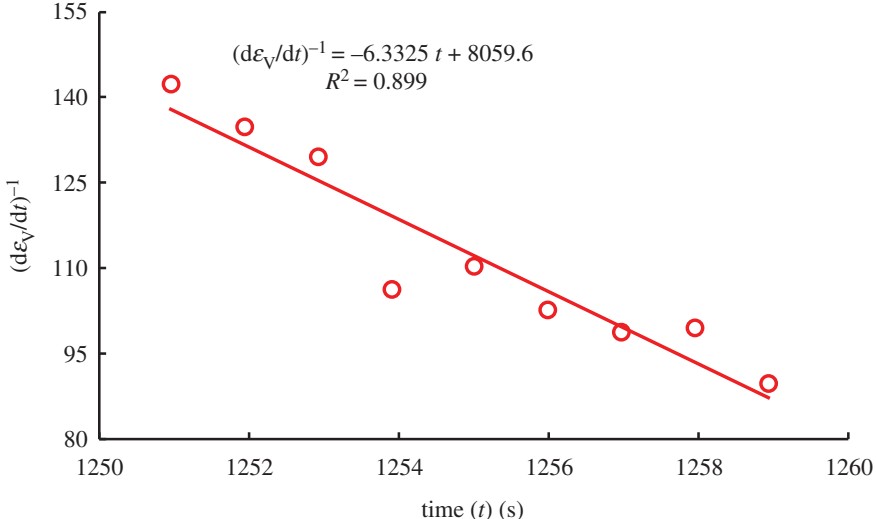

**Figure 10.** Short-term forecast for accelerated dilatancy.

**Table 2.** Duration of stress plateau.

| | $v$ (MPa s$^{-1}$) | | |
|---|---|---|---|
| | 0.02 | 0.05 | 0.1 |
| $t_{sp-15}$ (s) | 413.026 | 148.047 | 75.057 |
| $t_{sp-20}$ (s) | 370.016 | 219.130 | 137.948 |
| $t_{sp-30}$ (s) | 359.047 | 171.966 | 100.994 |

where $t$ is the real-time moment of the experiment recorded by the experimental apparatus in s. $A$ is constant. $\chi$ is an index to measure the degree of nonlinear, usually $1 < \chi < 2$. $\Omega$ is related to precursory strain.

The volumetric strain ($\varepsilon_V$) can be measured, so $\varepsilon_V$ is used to characterize $\Omega$

$$\frac{d^2\varepsilon_V}{dt^2} = A\left(\frac{d\varepsilon_V}{dt}\right)^{\chi}. \tag{3.15}$$

From the late stage of steady dilatancy, crack growth is uncontrolled, at which time $\chi = 2$

$$\left(\frac{d\varepsilon_V}{dt}\right)^{-1} = \left(\frac{d\varepsilon_V}{dt}\right)_0^{-1} - A(t - t_0). \tag{3.16}$$

In the paper, the case of $\sigma_3 = 15$ MPa, $v = 0.02$ MPa s$^{-1}$ is taken as an example for analysis. When the volumetric strain rate is zero (($d\varepsilon_V/dt)^{-1} = 0$), the equation (3.16) intersects the abscissa in figure 10, indicating that the crack growth is uncontrolled at the time. The results show that $t_{Eq. (3.16)} = 1272.736$ s is no different from the experimental real time of 1271.951 s. It is proved that equations (3.14), (3.15) and (3.16) can also be used to predict the failure time of sandstone under the stress path of constant axial stress and unloading confining pressure, which is also of interest in predicting seismic activity. The application of the above equations is based on the commonness of rock failure and cumulative damage, just as the sound emitted by breaking chopsticks accords with the three laws of earthquakes [45], and all of them are self-similar.

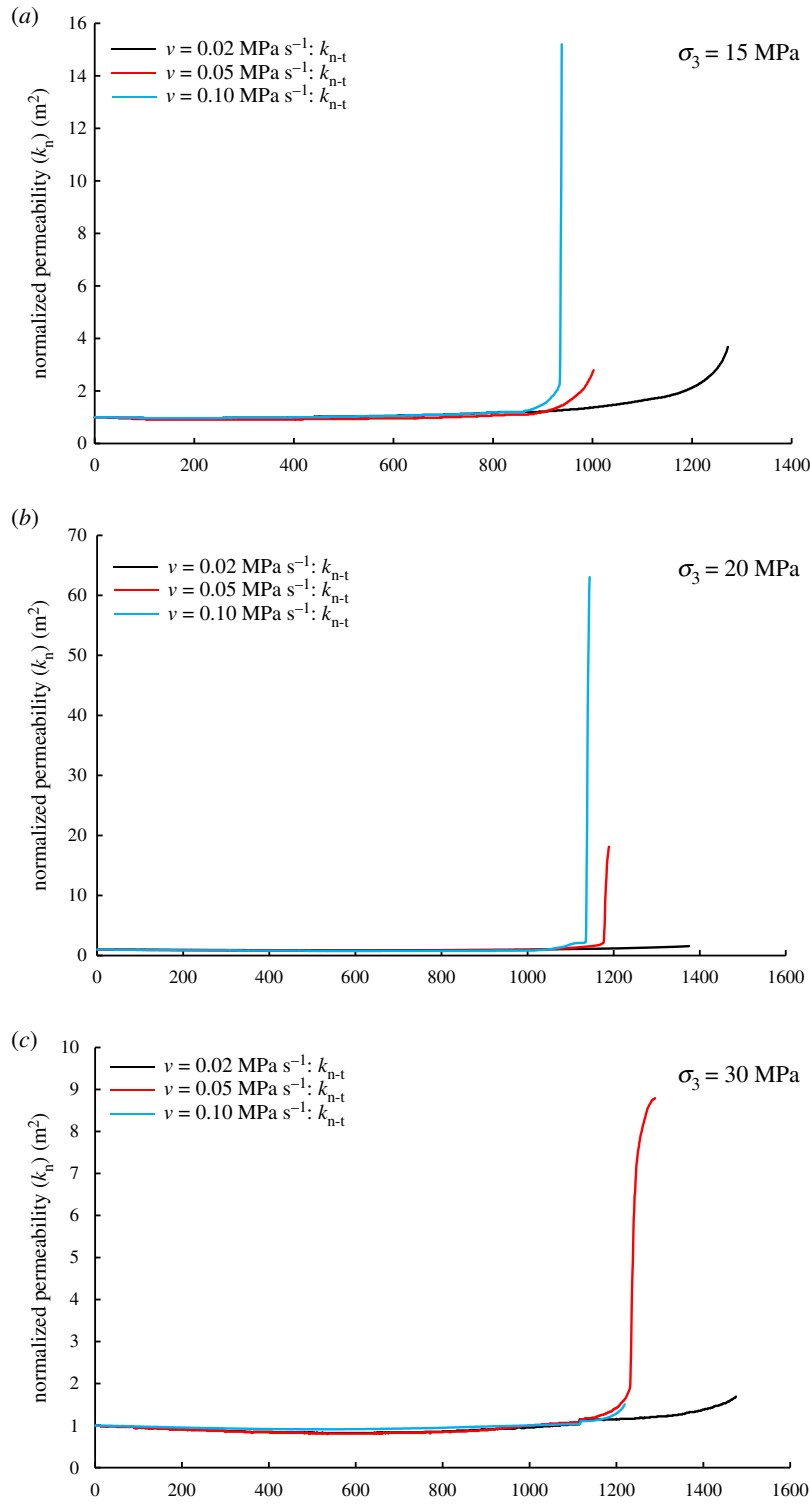

**Figure 11.** Normalized permeability ($k_n$) versus time ($t$) under different confining pressures and unloading rates. ($a$) $\sigma_3 = 15$ MPa, ($b$) $\sigma_3 = 20$ MPa and ($c$) $\sigma_3 = 30$ MPa.

## 4. Permeability evolution

The fracture system inside sandstone is relatively simple compared with the raw coal containing cleat system [46] and shale in which the gas slippage effect is more obvious [47]. The sandstone can be considered as an assembly of rock particles, and its permeability is usually governed by Darcy's law.

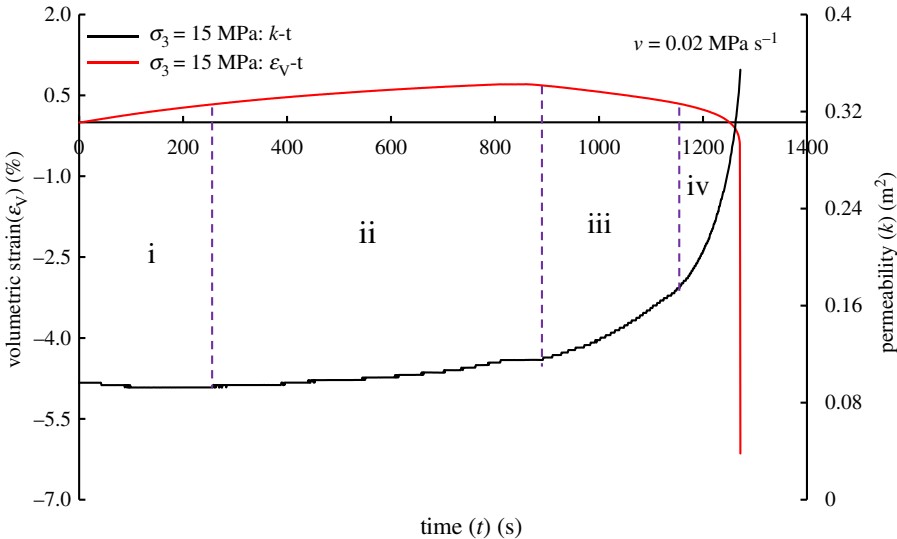

**Figure 12.** Volumetric strain ($\varepsilon_V$), normalized permeability ($k_n$) versus time ($t$) in the case of $\sigma_3 = 15$ MPa.

Therefore, the permeability equation is as follows [3,46]:

$$k = \frac{2q\mu L P_2}{A(P_1^2 - P_2^2)},\qquad(4.1)$$

where $k$ is the permeability in $m^2$. $q$ is the exit flow rate of $CH_4$ in $m^3\ s^{-1}$. $P_2$ is one standard atmospheric pressure in MPa. $\mu$ is the $CH_4$ kinematic viscosity of $CH_4$ in MPa s, at the temperature of the test according to Sutherland's formula. $L$ is the specimen length in m. $P_1$ is the entrance pressure in MPa of the test's $CH_4$ at the test temperature. $A$ is the cross-sectional area of the sandstone specimens in $m^2$.

There is an inevitable discreteness between the samples, resulting in different initial flow rates. Therefore, the permeability of sandstone is normalized, and its expression is as follows:

$$k_n = \frac{k_i}{k_0},\qquad(4.2)$$

where $k_0$ is the initial permeability at the beginning of unloading confining pressure in $m^2$. $k_i$ is the real-time permeability during unloading confining pressure in $m^2$. $k_n$ is the normalized permeability.

Figure 11 shows the normalized permeability ($k_n$) versus time ($t$) under different confining pressure and unloading rates. It can be seen that even under the experimental conditions of $\sigma_3 = 30$ MPa and $v = 0.05$ MPa $s^{-1}$, $k_n$ is the largest (figure 11c); however, $k_n$ is not the largest under the conditions of $v = 0.02$ MPa $s^{-1}$ and the three initial confining pressures, indicating that there is a positive correlation between the normalized permeability and the unloading rate. It also corresponds to the change of volumetric strain of sandstone before failure in table 1, i.e. the change of permeability is closely related to shear dilatancy.

Figure 12 shows the change of permeability during the experiment from loading axial stress (taking the initial confining pressure $\sigma_3 = 15$ MPa as an example). The change in permeability can be divided into four stages: *Permeability decreases*—the sandstone specimen is compressed, and the fracture is closed, resulting in the decrease of seepage channels; *Permeability increases slowly*—the sandstone specimen is still in a compressed state; however, with the further loading of the applied stress, the crack initiates and propagates, the tensile crack at the edge of the pre-existing inclined crack originates and gradually connects to the end of the adjacent inclined crack [21,22], and the fluid migration channel increases. *Permeability increases steadily*—the sandstone specimen enters the steady dilatancy stage and the inelastic volumetric strain can no longer be neglected [30]. *Permeability increases quickly*—with the further unloading of confining pressure, the rock continues to rebound, and the stress at the end of the fracture is highly concentrated which leads to local failure, prominent shear dilatancy and further opening and propagating of cracks. The specimen enters the accelerated dilatancy stage.

Figure 13 shows the variation of normalized permeability with volumetric strain in the unloading confining pressure stage. When the rate is low, the accelerated dilatancy (figure 13b) does not bring about the accelerated growth of permeability. The reason may be that the response of the internal fracture structure to the applied stress can be adjusted in time due to the small rate, which makes it

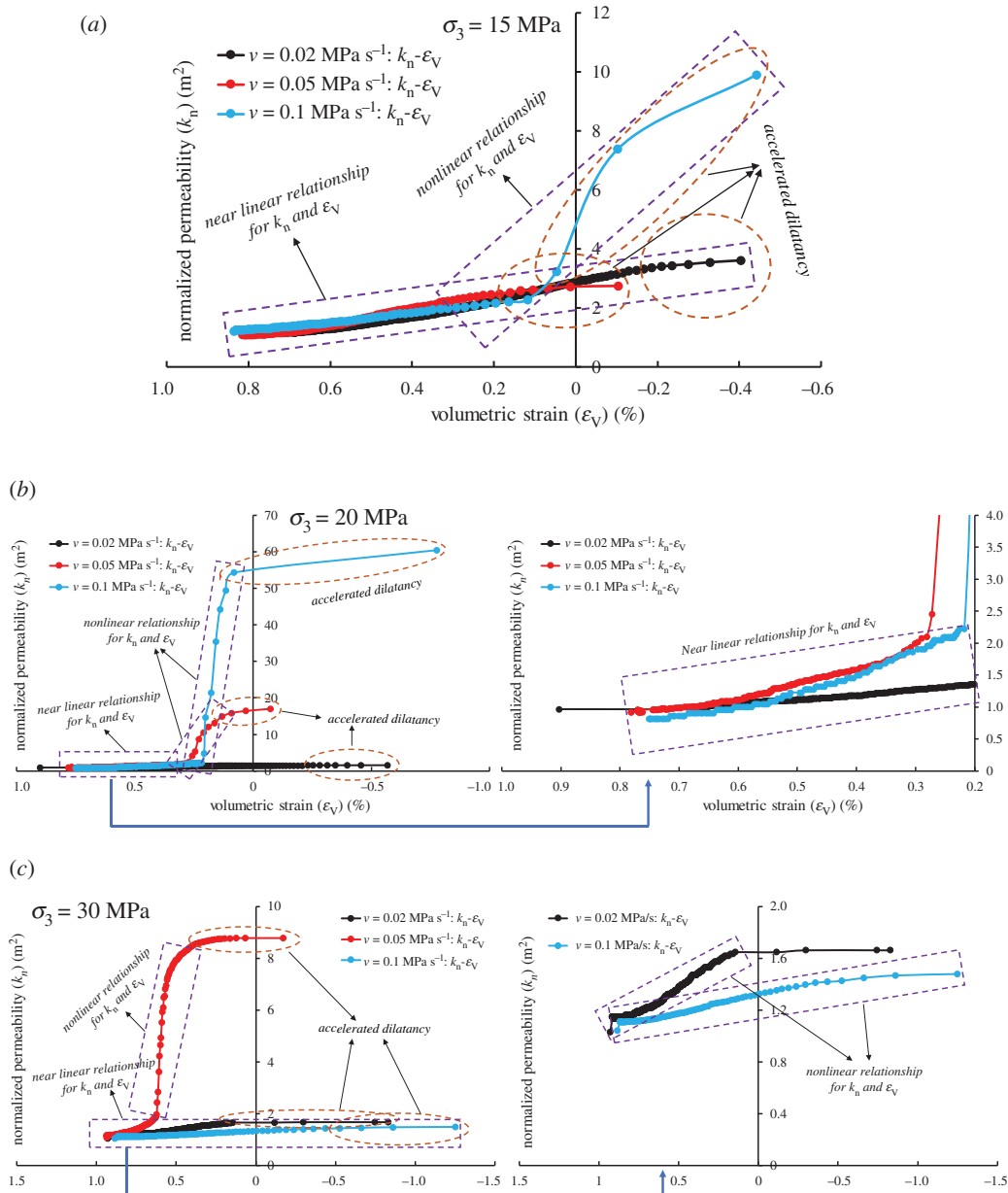

**Figure 13.** Normalized permeability ($k_n$) versus volumetric strain ($\varepsilon_V$) under different confining pressures and unloading rates. (a) $\sigma_3 = 15$ MPa, (b) $\sigma_3 = 20$ MPa and (c) $\sigma_3 = 30$ MPa.

easier to form dominant fractures, and the steadier fluid flow. In the steady dilatancy stage, the normalized permeability is linear with the volumetric strain. In the accelerated dilatancy stage, the two are linear at a low rate and nonlinear at a high rate (such as a polynomial relationship [48], or power law relationship [49]). The linear/nonlinear relationship between normalized permeability and volumetric strain can directly reflect the temporal characteristics of mutation (table 3), which is of interest to us. The feature may help us to effectively prevent engineering hazards, such as shear failure induced by local shear strain on the slope [50], as well as coal and gas outburst [3].

By analysing the experimental results (figure 13 and table 3), we summarize figure 14 to provide the theoretical basis for tunnel and underground roadway excavation. Taking the initial confining pressure $\sigma_3 = 20$ MPa as an example, it can be seen from table 3 and figure 14 that there is no mutation in volumetric strain and normalized permeability when $v = 0.02$ MPa s$^{-1}$. Only in the period from $t = 1373.089$ to $1374.075$ s, the volumetric strain has changed significantly. However, the change is noteworthy. Under the experimental conditions or specific engineering conditions, it is necessary to observe the surrounding rock deformation to prevent engineering geological hazards, especially in the

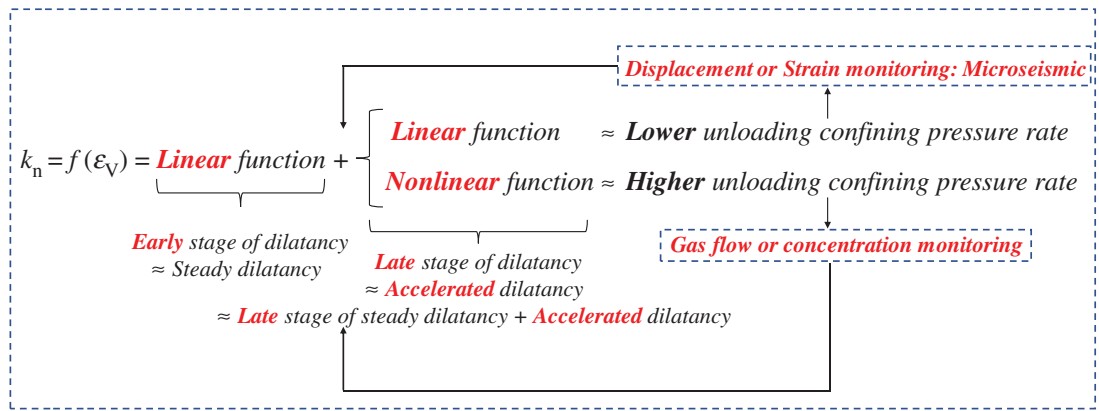

**Figure 14.** Generalization of the relationship between volumetric strain and permeability under different unloading rates and its enlightenment for the prevention and control of unloading engineering disasters.

**Table 3.** Temporal characteristics of volumetric strain mutation and normalized permeability of sandstone in a few seconds of adjacent failure. Note that $t$ here is the real-time moment of the experiment recorded by the experimental apparatus.

| | | $\sigma_3 = 20$ MPa | | | |
|---|---|---|---|---|---|
| $v = 0.02$ MPa s$^{-1}$ | $t$ (s) | 1371.014 | 1372.104 | 1373.089 | 1374.075 |
| | $\varepsilon_V$ (%) | −0.348 | −0.398 | −0.455 | −0.569 |
| | $k_n$ | 1.545 | 1.546 | 1.546 | 1.558 |
| $v = 0.05$ MPa s$^{-1}$ | $t$ (s) | 1176.204 | 1177.192 | 1178.173 | 1179.161 |
| | $\varepsilon_V$ (%) | 0.281 | 0.272 | 0.258 | 0.245 |
| | $k_n$ | 2.097 | 2.454 | 4.135 | 5.402 |
| $v = 0.1$ MPa s$^{-1}$ | $t$ (s) | 1135.137 | 1136.127 | 1137.216 | 1138.198 |
| | $\varepsilon_V$ (%) | 0.217 | 0.207 | 0.201 | 0.177 |
| | $k_n$ | 2.227 | 4.773 | 14.636 | 21.409 |

vicinity of the working face where there are faults. In the case of faults, strain monitoring should be strengthened to prevent the angle between the maximum compression direction and the fracture plane from being within the activation range, resulting in the reactivation of faults and natural fracture populations [11]. In addition to the specific cross-section, strain acquisition is mainly obtained by indirect measurement (strain factor $\propto$ energy$^{1/2}$) [51,52], so it is necessary to focus on monitoring the microseism swarm events near the working face [43].

However, when $v = 0.1$ MPa s$^{-1}$, the normalized permeability undergoes changes significantly earlier than the volumetric strain during the period $t = 1135.137–1137.216$ s, and the volumetric strain changes are relatively steady, i.e. the rock is in a steady dilatancy stage. It indicates that the prevention of engineering geological hazards should be focused on observing the change of gas flow or concentration (figure 14) under such experimental conditions or some engineering conditions. The elliptical frame in figure 13 represents the accelerated dilatancy of sandstone during unloading confining pressure, and the rectangular frame represents the linear or nonlinear relationship between normalized permeability and volumetric strain.

It should be noted that the two monitoring purposes mentioned in figure 14 should be coordinated with each other, but with different emphasis. In practice, stress monitoring should run through the entire engineering activity, and both stress and strain can drive the tectonic effect [12]. Furthermore, the core issue is to extend the time of mutation occurring or to completely address the potential major hazards. We also encounter an exceedingly crucial engineering problem, i.e. the manner in which the unloading rate can be quantitatively or qualitatively distinguished as high or low. In the paper, it has not been studied, but the characteristics of fracture development on rock wall, such as spalling,

layered crack [2], can be observed from the tunnel face and mining working face. This allows a comprehensive judgement to be made.

## 5. Conclusion

Triaxial experiments of sandstones subjected to different initial confining pressures and unloading rates were performed to investigate the deformation behaviour and permeability evolution of sandstone under the engineering conditions related to the unloading path. The main conclusions are summarized as follows:

(1) Unloading stress state reduced the normal stress acting on the fracture and made the slip along the fracture more likely to occur. Sandstone samples experienced shear dilatancy before the failure, and the greater the deviator stress at the sandstone failure, the more obvious the shearing effect. It was fully proved that there were local sandstone particles dispersing on the macroscopic fracture surface, which means the shear spalling occurred.

(2) Dilatancy was not suppressed with an increase of initial confining pressure, i.e. the dilatancy factor did not decrease, and it showed that the ratio of the plastic volumetric strain increment to plastic axial strain increment was larger. Correspondingly, the crack density parameter increased with the increase in the initial unloading confining pressure, which is consistent with the change of volumetric strain. In addition, normalized permeability was positively correlated with the unloading rates. The permeability was closely related to the shear dilatancy behaviour.

(3) Sandstone deformation experienced two stages in unloading stress paths: the stress platform stage and the failure instability stage. Sandstone failure was also the result of cumulative damage, and the concept of time-to-failure can be used to predict the failure time during the instability stage.

(4) Linear/nonlinear relationship associated with the unloading rate between normalized permeability and volumetric strain can directly reflect the temporal characteristics of the mutation of the two. If the mutation of permeability is earlier than that of volumetric strain, we should focus on monitoring gas flow or concentration changes to prevent engineering geological hazards, and if the mutation of volumetric strain is earlier than that of permeability, it is necessary to focus on monitoring the microseism swarm events near the working face.

Data accessibility. The datasets supporting this article have been uploaded as part of the electronic supplementary material.

Authors' contributions. H.Z. carried out the tests, participated in data analysis, the design of the study and drafted the manuscript. C.L. carried out the statistical analyses and critically revised the manuscript, and conceived the original idea for this study. G.H. coordinated the study and helped draft the manuscript. All authors gave final approval for publication and agree to be held accountable for the work performed therein.

Competing interests. We declare we have no competing interests.

Funding. This study was financially supported by the National Natural Science Foundation of China (grant no. 51674049) and the Graduate Research and Innovation Foundation of Chongqing (grant nos. CYB19046 and CYB19045).

Acknowledgements. We appreciate Beichen Yu for supporting the experiment of our study.

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
