## [Reviewer comments · Royal Society Open Science]

Review History

RSOS-201792.R0 (Original submission)

Review form: Reviewer 1

Is the manuscript scientifically sound in its present form?

Yes

Are the interpretations and conclusions justified by the results?

Yes

Is the language acceptable?

Yes

Do you have any ethical concerns with this paper?

No

Have you any concerns about statistical analyses in this paper?

Yes

Recommendation?

Accept with minor revision (please list in comments)

Comments to the Author(s)

The topic is interesting. Mechanical response, deformation behavior and permeability evolution of surrounding rock under unloading conditions are of significant importance in rock engineering activities. The triaxial mechanics and seepage experiments of sandstone with different combinations of initial confining pressure and unloading confining pressure rate under fixed axial stress are performed in order to investigate the dilatancy behavior, permeability evolution, and their relationship. It provides us a better understanding of the dilatancy behavior and its impact on permeability evolution.

Only minor revision is needed.

1. Page 4: It is suggested to reorganize Fig.1 to provide more information.
2. Page 6, Page 7: Please check the unit in Fig. 4.
3. Please check the conclusion "Dilatancy is not suppressed with an increase of initial confining pressure."

Review form: Reviewer 2**Is the manuscript scientifically sound in its present form?**

Yes

Are the interpretations and conclusions justified by the results?

Yes

Is the language acceptable?

Yes

Do you have any ethical concerns with this paper?

No

Have you any concerns about statistical analyses in this paper?

No

Recommendation?

Accept with minor revision (please list in comments)

Comments to the Author(s)

The manuscript presented an interesting investigation on dilatancy behavior and permeability evolution under different unloading rates. The manuscript is well written. The approach taken and result are interesting and useful for potential readers. This study is helpful to better understand the mechanical behaviors and permeability characteristics of rock during the excavation. Specific comments for revision are as follows,

- (1) In abstract, line 15, the tenses of the two sentences are inconsistent. Please check and revise them.
- (2) Please give the schematic diagram of the experimental scheme.
- (3) In Fig. 3, the strengths of sandstones without gas pressure are given. However, these strengths are not used in the paper, please confirm it.

- (4) In section 3.1, line 187, the author uses the dilatancy factor (DF) to characterize the shear localization behavior. However, the content of the following section is about the crack density parameter until line 229. Please rearrange the logical relationship between the paragraphs.
- (5) There are some grammatical errors in this paper. For example, "the shear-dilatancy induced failure of the surrounding rock of the tunnel comes from the unloading dilatancy" (line 241) and "We focus on if the dilatancy behavior is significantly suppressed by the increase of initial confining pressure during unloading." (line 248).
- (6) In line 250, "The deviator stress-axial strain fitting represented by Fig. 7 (b) is performed in the unloading confining pressure process". Fig. 7 contains two figures. What is the role of Fig. 7(a)?
- (7) In line 427, "By analyzing the experimental results (Fig. 12 and Table 3), we summarize Fig. 13 to provide the theoretical basis for tunnel and underground roadway excavation". However, the author does not explain the meaning of Fig. 13 in the following section.

Decision letter (RSOS-201792.R0)

Dear Dr Liu

On behalf of the Editors, we are pleased to inform you that your Manuscript RSOS-201792 "Dilatancy behavior and permeability evolution of sandstone subjected to initial confining pressures and unloading rates" has been accepted for publication in Royal Society Open Science subject to minor revision in accordance with the referees' reports. Please find the referees' comments along with any feedback from the Editors below my signature.

Please submit your revised manuscript and required files (see below) no later than 7 days from today's (ie 25-Nov-2020) date. Note: the ScholarOne system will 'lock' if submission of the revision is attempted 7 or more days after the deadline. If you do not think you will be able to meet this deadline please contact the editorial office immediately.

on behalf of Dr Philip Benson (Associate Editor) and R. Kerry Rowe (Subject Editor)
 openscience@royalsociety.org

Reviewer comments to Author:

Reviewer: 1

Comments to the Author(s)

The topic is interesting. Mechanical response, deformation behavior and permeability evolution of surrounding rock under unloading conditions are of significant importance in rock engineering activities. The triaxial mechanics and seepage experiments of sandstone with different combinations of initial confining pressure and unloading confining pressure rate under fixed axial stress are performed in order to investigate the dilatancy behavior, permeability evolution, and their relationship. It provides us a better understanding of the dilatancy behavior and its impact on permeability evolution.

Only minor revision is needed.

1. Page 4: It is suggested to reorganize Fig.1 to provide more information.
2. Page 6, Page 7: Please check the unit in Fig. 4.
3. Please check the conclusion "Dilatancy is not suppressed with an increase of initial confining pressure."

Reviewer: 2

Comments to the Author(s)

The manuscript presented an interesting investigation on dilatancy behavior and permeability evolution under different unloading rates. The manuscript is well written. The approach taken and result are interesting and useful for potential readers. This study is helpful to better understand the mechanical behaviors and permeability characteristics of rock during the excavation. Specific comments for revision are as follows,

- (1) In abstract, line 15, the tenses of the two sentences are inconsistent. Please check and revise them.
- (2) Please give the schematic diagram of the experimental scheme.
- (3) In Fig. 3, the strengths of sandstones without gas pressure are given. However, these strengths are not used in the paper, please confirm it.
- (4) In section 3.1, line 187, the author uses the dilatancy factor (DF) to characterize the shear localization behavior. However, the content of the following section is about the crack density parameter until line 229. Please rearrange the logical relationship between the paragraphs.
- (5) There are some grammatical errors in this paper. For example, "the shear-dilatancy induced failure of the surrounding rock of the tunnel comes from the unloading dilatancy" (line 241) and "We focus on if the dilatancy behavior is significantly suppressed by the increase of initial confining pressure during unloading." (line 248).
- (6) In line 250, "The deviator stress-axial strain fitting represented by Fig. 7 (b) is performed in the unloading confining pressure process". Fig. 7 contains two figures. What is the role of Fig. 7(a)?
- (7) In line 427, "By analyzing the experimental results (Fig. 12 and Table 3), we summarize Fig. 13 to provide the theoretical basis for tunnel and underground roadway excavation". However, the author does not explain the meaning of Fig. 13 in the following section.

===PREPARING YOUR MANUSCRIPT===

===PREPARING YOUR REVISION IN SCHOLARONE===

- If you are providing image files for potential cover images, please upload these at this step, and inform the editorial office you have done so. You must hold the copyright to any image provided.
- A copy of your point-by-point response to referees and Editors. This will expedite the preparation of your proof.

- Ensure that your data access statement meets the requirements at <https://royalsociety.org/journals/authors/author-guidelines/#data>. You should ensure that you cite the dataset in your reference list. If you have deposited data etc in the Dryad repository, please only include the 'For publication' link at this stage. You should remove the 'For review' link.
- If you are requesting an article processing charge waiver, you must select the relevant waiver option (if requesting a discretionary waiver, the form should have been uploaded at Step 3 'File upload' above).
- If you have uploaded ESM files, please ensure you follow the guidance at <https://royalsociety.org/journals/authors/author-guidelines/#supplementary-material> to include a suitable title and informative caption. An example of appropriate titling and captioning may be found at https://figshare.com/articles/Table_S2_from_Is_there_a_trade-off_between_peak_performance_and_performance_breadth_across_temperatures_for_aerobic_scooping_in_teleost_fishes_/3843624.

Author's Response to Decision Letter for (RSOS-201792.R0)

See Appendix A.

Decision letter (RSOS-201792.R1)

Dear Dr Liu,

It is a pleasure to accept your manuscript entitled "Dilatancy behavior and permeability evolution of sandstone subjected to initial confining pressures and unloading rates" in its current form for publication in Royal Society Open Science.

Due to rapid publication and an extremely tight schedule, if comments are not received, your paper may experience a delay in publication. Royal Society Open Science operates under a

continuous publication model. Your article will be published straight into the next open issue and this will be the final version of the paper. As such, it can be cited immediately by other researchers. As the issue version of your paper will be the only version to be published I would advise you to check your proofs thoroughly as changes cannot be made once the paper is published.

on behalf of Dr Philip Benson (Associate Editor) and R. Kerry Rowe (Subject Editor)
openscience@royalsociety.org

Appendix A

Detailed Responses to Editors and Reviewers

Manuscript Number: RSOS-201792

Title: Dilatancy behavior and permeability evolution of sandstone subjected to initial confining pressures and unloading rates

Authors: Honggang Zhao, Chao Liu, Gun Huang

Dear editors and reviewers,

Special thanks to the anonymous reviewers for their careful reviews and valuable comments. We have carefully checked these comments and made necessary corrections and modifications. We tried our best to improve the manuscript.

Improvements and clarifications have been made in the current manuscript. These changes are marked by **red words**. English grammar errors and inaccurate expressions are modified and rephrased.

We appreciate for the editors and reviewers' warm work earnestly, and hope that the manuscript after the corrections can satisfy the requirements for publication. Once again, thank you very much for your comments and suggestions.

Point-by-point Responses to the editors' and reviewers' comments

Reviewer #1:

Comment 1: Page 4: It is suggested to reorganize Fig.1 to provide more information.

Response:

Thank you so much for this constructive comment and suggestion. In order to show the experimental equipment more clearly, we add the schematic diagram of apparatus and annotate the name of device in the physical diagram, as shown in Fig. 1.

Fig. 1. Schematic and physical diagram of the apparatus

Comment 2: Page 6, Page 7: Please check the unit in Fig. 4.

Response:

Thank you for this comment and I'm so sorry that my carelessness has brought you difficulties in reviewing. The unit in Fig. 4 has been corrected, as shown in Fig. 4.

Thanks again for your comment, and we apologize for our carelessness.

(a)

Fig. 4. Stress-strain relationship of sandstone under different unloading rates. (a) $v=0.02$ MPa/s. (b) $v=0.05$ MPa/s. (c) $v=0.1$ MPa/s.

Comment 3: Please check the conclusion “Dilatancy is not suppressed with an increase of initial confining pressure”.

Response:

Thank you for this comment. Based on the experimental results (Table 1 in the manuscript), we found that the dilatancy factor (DF) did not decrease with the increase of confining pressure (line 209). The decrease of Df means that the shear dilatancy is suppressed. Thus, the conclusion that dilatancy is not suppressed with an increase in initial confining pressure is obtained. The main reason for this phenomenon is that the above phenomenon is related to the mechanical properties and stress state of sandstone. The strength of the sandstone used in our research is greater than that in related literature ^[1, 2], and the experimental conditions do not meet the stress regime of the brittle-ductile transition of rock, and there is no cataclastic flow ^[2, 3]. In addition, we studied the dilatancy behavior of sandstone under the stress path of unloading confining pressure, rather than the loading axial stress ^[1, 2]. There are significant differences in the deformation behavior of sandstone caused by the stress path between the two. For example, the failure of the surrounding rock of the tunnel induced by shear-dilatancy comes from the unloading dilatancy, and the dilatancy failure in triaxial compression test comes from loading failure. The inducement of dilatancy behavior is different

between the two. In a brittle state, high confining pressure is conducive to fracture compaction, meaning natural or pre-existing closed cracks increase. Closed cracks are the necessary condition for shear sliding [4], and coupled with high deviator stress, resulting in that the shear dilatancy of sandstone is significant under $\sigma_3=30$ MPa.

Reviewer #2:

Comment 1: In abstract, line 15, the tenses of the two sentences are inconsistent. Please check and revise them.

Response:

Thank you for your carefully reviewing. As Reviewer suggested that we have revised this error and checked the whole manuscript. The modified parts are marked in red words. Thank you again for your careful review.

Comment 2: Please give the schematic diagram of the experimental scheme.

Response:

Thank you for this comment. In this paper, two kinds of tests are carried out, i.e. conventional triaxial compression test under different confining pressures and unloading confining pressure test under different unloading rates. Thus, there are two stress paths diagram, as shown in Fig. 3.

Fig. 3. Schematic diagram of the experimental scheme: (a) conventional triaxial compression test; (b) unloading confining pressure test

Comment 3: In Fig. 3, the strengths of sandstones without gas pressure are given. However, these strengths are not used in the paper, please confirm it.

Response:

Thank you very much for this comment. The purpose of giving the strengths of sandstone without gas pressure is to compare the strength difference between sandstones with gas pressure and non-gas pressure. We have deleted the strengths of sandstone without gas pressure in the revised manuscript. The modified figure is shown as follows:

Fig. 3. Relationship between maximum deviator stress and confining pressure under conventional triaxial compression test with 3 MPa gas pressure.

Comment 4: In section 3.1, line 187, the author uses the dilatancy factor (DF) to characterize the shear localization behavior. However, the content of the following section is about the crack density parameter until line 229. Please rearrange the logical relationship between the paragraphs.

Response:

Thank you for your carefully reviewing. We have rearranged the position of the paragraph, and put the content about the dilatancy factor (DF) together. So that the logical relationship between the content of the paper is closely linked. The modified parts are marked in red words in Section 3.1.

Comment 5: There are some grammatical errors in this paper. For example, “the shear-dilatancy induced failure of the surrounding rock of the tunnel comes from the unloading dilatancy” (line 241) and “We focus on if the dilatancy behavior is

significantly suppressed by the increase of initial confining pressure during unloading.” (line 248).

Response:

Thank you for this comment. We have checked the manuscript and corrected grammatical errors and spelling mistakes and so on. The modified parts are marked in red words. The grammatical errors mentioned by the reviewer #2 are revised as follows: “the failure of the surrounding rock of the tunnel induced by shear-dilatancy comes from the unloading dilatancy” and “We focus on whether the dilatancy behavior is significantly suppressed by the increase of initial confining pressure during unloading process.”.

Comment 6: In line 250, “The deviator stress-axial strain fitting represented by Fig. 7 (b) is performed in the unloading confining pressure process”. Fig. 7 contains two figures. What is the role of Fig. 7(a)?

Response:

Thank you for this comment. Fig. 7(a) shows the stress-time and strain-time curves of the whole process from loading to unloading failure, and Fig. 7(b) is a partial enlarged view of Fig. 7(a). Fig. 7(a) is mainly used to illustrate the content which is mentioned in Section 3.2, i.e., the failure of sandstone occurs only after the axial stress reaches the target value for a period of time, and then failure is instantaneous. The failure process of sandstone can be regarded as two stages: the stress plateau stage and the failure instability stage.

Comment 7: In line 427, “By analyzing the experimental results (Fig. 12 and Table 3), we summarize Fig. 13 to provide the theoretical basis for tunnel and underground roadway excavation”. However, the author does not explain the meaning of Fig. 13 in the following section.

Response:

Thank you for this comment. It should be noted that since the stress path diagram was added in the manuscript (Fig. 3), thus Fig. 13 is renamed as Fig. 14. In order to

make the logic clearer, we put the sentence “By analyzing the experimental results (Fig. 13 and Table 3), we summarize Fig. 14 to provide the theoretical basis for tunnel and underground roadway excavation.” in line 402. The explanation of Figure 14 can be seen in line 413 to 433, i.e. Taking the initial confining pressure $\sigma_3=20$ MPa as an example, it can be seen from Table 3 and Fig. 14 that there is no mutation in volumetric strain and normalized permeability when $\nu = 0.02$ MPa/s. Only in the period from $t=1373.089$ s to 1374.075 s, the volumetric strain has changed significantly. However, the change is noteworthy. Under the experimental conditions or specific engineering conditions, it is necessary to observe the surrounding rock deformation to prevent engineering geological hazards, especially in the vicinity of the working face where there are faults. In the case of faults, strain monitoring should be strengthened to prevent the angle between the maximum compression direction and the fracture plane from being within the activation range, resulting in the reactivation of faults and natural fracture populations [5]. In addition to the specific cross-section, strain acquisition is mainly obtained by indirect measurement (strain factor \propto energy^{1/2}) [6, 7], so it is necessary to focus on monitoring the microseisms swarm events near the working face [8].

However, when $\nu=0.1$ MPa/s, the normalized permeability undergoes changes significantly earlier than the volumetric strain during the period $t=1135.137$ s to 1137.216 s, and the volumetric strain changes are relatively steady, i.e. the rock is in a steady dilatancy stage. It indicates that the prevention of engineering geological hazards should be focused on observing the change of gas flow or concentration (Fig. 14) under such experimental conditions or some engineering conditions. The elliptical frame in Fig. 13 represents the accelerated dilatancy of sandstone during unloading confining pressure, and the rectangular frame represents the linear or nonlinear relationship between normalized permeability and volumetric strain. We apologize for our inattention.

References

[1] Ougier A, Zhu W. Effect of pore pressure buildup on slowness of rupture

propagation. *Journal of Geophysical Research: Solid Earth* 2015; 120(12):7966–85.
<https://doi.org/10.1002/2015JB012047>

[2] Wong T, David C, Zhu W. The transition from brittle faulting to cataclastic flow in porous sandstones: Mechanical deformation. *Journal of Geophysical Research: Solid Earth* 1997; 102(B2):3009–25. <https://doi.org/10.1029/96JB03281>

[3] Ougier-Simonin A, Zhu W. Effects of pore fluid pressure on slip behaviors: An experimental study. *Geophysical Research Letters* 2013; 40(11):2619–24.
<https://doi.org/10.1002/2015JB012047>

[4] David EC, Brantut N, Schubnel A, Zimmerman RW. Sliding crack model for nonlinearity and hysteresis in the uniaxial stress–strain curve of rock. *International Journal of Rock Mechanics and Mining Sciences* 2012; 52:9–17.
<https://doi.org/10.1016/j.ijrmms.2012.02.001>

[5] Jaeger JC, Cook NG, Zimmerman R. *Fundamentals of Rock Mechanics*, 3rd, ed., Chapman Hall, London 2007.

[6] Ying W, Benson PM, Young RP. Laboratory simulation of fluid-driven seismic sequences in shallow crustal conditions. *Geophysical Research Letters* 2009; 36(20).
<https://doi.org/10.1029/2009GL040230>

[7] Argus DF, Lyzenga GA. Site velocities before and after the Loma Prieta and Gulf of Alaska earthquakes determined from VLBI. *Geophysical Research Letters* 1994; 21(5):333–6. <https://doi.org/10.1029/94GL00027>

[8] Lee MK, Wolf LW. Analysis of fluid pressure propagation in heterogeneous rocks: Implications for hydrologically-induced earthquakes. *Geophysical Research Letters* 1998; 25(13):2329–32. <https://doi.org/10.1029/98GL01694>

Best regards.

Chao Liu

Corresponding author

E-mail: liuc_rm@cumt.edu.cn